# Immunity against Non-Melanoma Skin Cancer and the Effect of Immunosuppressive Medication on Non-Melanoma Skin Cancer Risk in Solid Organ Transplant Recipients

**DOI:** 10.3390/cells12202441

**Published:** 2023-10-11

**Authors:** Dixie Bakker, Walbert J. Bakker, Marcel W. Bekkenk, Rosalie M. Luiten

**Affiliations:** 1Department of Dermatology, Netherlands Institute for Pigment Disorders, Amsterdam University Medical Centers, University of Amsterdam, 1105 AZ Amsterdam, The Netherlands; 2Cancer Center Amsterdam, Amsterdam Institute for Infection and Immunity, 1081 HV Amsterdam, The Netherlands; 3Amsterdam University Medical Centers, VU University of Amsterdam, 1081 HV Amsterdam, The Netherlands

**Keywords:** non-melanoma skin cancer, cancer immunity, transplantation, immune suppression

## Abstract

Non-melanoma skin cancers (NMSCs) occur frequently in the Caucasian population and are considered a burden for health care. Risk factors include ultraviolet (UV) radiation, ethnicity and immunosuppression. The incidence of NMSC is significantly higher in solid organ transplant recipients (SOTRs) than in immunocompetent individuals, due to immunosuppressive medication use by SOTRs. While the immunosuppressive agents, calcineurin inhibitors and purine analogues increase the incidence of NMSC in transplant recipients, mTOR inhibitors do not. This is most likely due to the different immunological pathways that are inhibited by each class of drug. This review will focus on what is currently known about the immune response against cutaneous squamous cell carcinoma (cSCC) and basal cell carcinoma (BCC), two of the main types of NMSC. Furthermore, we will describe the different classes of immunosuppressants given to SOTRs, which part of the immune system they target and how they can contribute to NMSC development. The risk of developing NMSC in SOTRs is the result of a combination of inhibiting immunological pathways involved in immunosurveillance against NMSC and the direct (pro/anti) tumor effects of immunosuppressants.

## 1. Introduction

Non-melanoma skin cancers (NMSCs), including basal cell carcinoma (BCC) and cutaneous squamous cell carcinoma (cSCC), are the most frequent types of cancers among Caucasians. While overall mortality rates of these cancers are low, the steadily increasing number of cases continues to cause burden to society and health care. Risk factors for developing NMSC include chronic exposure to ultraviolet radiation (UVR), age, ethnicity and family history of skin cancer [1]. More than 90% of BCC show the upregulation of the Hedgehog pathway, involving the Patched 1 (PTCH1) gene and smoothened (SMO) and glioma-associated (GLI) oncogenes. The Hedgehog pathway is implicated in self-renewal, survival and angiogenesis and can drive a cancer stem phenotype. Although BCC rarely metastasize, local invasion and tissue destruction cause high morbidity. In cSCC, a more diverse (UV-induced) gene mutational profile has been found, including tumor suppressors, epigenetic regulators and DNA repair pathways, leading to a high neoantigen burden [1]. The mortality of cSCC is due to its metastasizing behavior to distant sites.

Immunosuppression is another major risk factor, as demonstrated by the increased incidence rate among HIV patients and solid organ transplant recipients (SOTRs) undergoing a long-term immunosuppressive regimen [2,3]. This indicates the importance of immunosurveillance in preventing NMSC. However, the immunological mechanisms mediating immunosurveillance against NMSC are not fully known. Interestingly, the risk of NMSC in SOTRs depends on the type of immunosuppressive medication. While calcineurin inhibitors increase the risk of NMSC development, mTOR inhibitors have been shown to reduce the risk of NMSC development [4].

In this review, we discuss the current knowledge of immunity against NMSC (Figure 1). Moreover, we analyze the targets of immunosuppressive medication that increase NMSC risk to identify the immunological mechanisms involved in NMSC immunosurveillance.

## 2. Immunity against NMSC

### 2.1. Innate Immunity against NMSC

#### 2.1.1. Dendritic Cells

Dendritic cells residing in the skin have a central role in detecting aberrant keratinocytes and, upon activation and migration to the lymph node, prime adaptive immune responses (Figure 1). Dendritic cells are considered to act as a bridge between the innate and adaptive immune system [5]. Functionally, there are two main types of dendritic cells: conventional dendritic cells (cDCs) and plasmacytoid dendritic cells (pDCs). Although pDCs are usually not found in normal skin, they produce type I interferons, such as IFNα, which play an important role in antitumor immunity. Several studies have shown that intralesional injection with IFN-⍺, a type I interferon, can be an effective treatment for both cSCC and BCC [6,7]. Furthermore, the expression of type I interferon signaling protein ISFG-3 is suppressed in early skin carcinogenesis, suggesting a role for these cytokines during cancer development [8].

Skin DCs comprise Langerhans cells (LCs) and cDCs and, in the skin, are referred to as dermal dendritic cells (dDCs). The role of skin DCs in NMSC has been shown in studies of UV-induced chronic inflammation and immune suppression in the skin, which both contribute to NMSC development [9]. UVB induces the apoptosis of epidermal skin cells, predominantly keratinocytes, and cutaneous inflammation. In a mouse model, UVB-induced inflammation was exacerbated and prolonged when LCs were depleted, coinciding with an increased number of apoptotic cells. LCs were shown to phagocytose apoptotic keratinocytes upon UVB exposure, and thereby play an essential role in the resolution of UVB-induced skin inflammation [10]. The role of LCs in skin cancer surveillance was also shown in a murine model of chemical skin carcinogenesis [11]. Initiation and early events during carcinogenesis induced epidermal LCs in order to secrete TNF⍺, resulting in the recruitment of NK cells to the epidermis and prevention of cSCC development. Conversely, the depletion of both LCs and NK cells resulted in increased tumor growth [11]. Regarding UV-induced immunosuppression, Furio et al. (2005) [12] reported that the UVA irradiation of ex vivo human skin resulted in a decreased number of DC emigrating from the skin and showing impaired DC maturation. These findings suggest that even though surviving UVA-exposed dDCs can reach the lymph nodes, they exhibit impaired antigen presentation and might fail to induce an effective T cell response in vivo. In support of this, DCs in human cSCC tissue were shown to have impaired ability to stimulate T cell proliferation compared to DCs from healthy skin [13].

Altogether, these findings suggest that skin DCs play an important role in skin cancer immunosurveillance and in initiating adaptive immune responses against NMSC.

#### 2.1.2. Macrophages

Tumor-associated macrophages (TAMs) are found within and around cSCC and BCC as part of the tumor microenvironment (TME). Pro-inflammatory/M1 macrophages have antitumor properties and promote inflammation, whereas anti-inflammatory/M2 macrophages contribute to an immunosuppressive environment and decreased immunosurveillance that allows tumor growth (Figure 1) [14]. Under the influence of the TME, in most types of cancer, monocytes are stimulated to differentiate into M2 macrophages. Increased M2 macrophage infiltration is often associated with poor cancer prognosis [15,16]. TAMs with a M2 phenotype promote a Type 2 T cell response (Th2) and attract immunosuppressive T regulatory cells (Tregs), resulting in an inefficient antitumor response. This type of immune response also stimulates differentiation of TAMs into M2 macrophages as a positive feedback loop [17]. However, Pettersen et al. (2011) [18] reported that not M2, but M1 gene sets were enriched in cSCC compared to normal skin, indicating the activation of M1 macrophages and the promotion of a Type 1 T cell response (Th1). In addition, several M2-specific genes were upregulated and the TAMs also expressed Th2-associated products. This suggests that the composition of macrophages in the TME is heterogeneous and consists of both M1 and M2 macrophages and a strong Th1 response. TAMs can promote carcinogenesis in cSCC by releasing cytokines, including vascular endothelial growth factor-C (VEGF-C) and matrix metalloproteinase (MMP) 9 and 11, which stimulate lymph-angiogenesis and tumor growth [18,19]. The expression of VEGFC in cSCC TME coincided with enhanced lymphatic density and reorganized lymphatic endothelial vessels in the peritumoral dermis, which may facilitate metastases [19]. In BCC, macrophages were found more abundantly in tumors that did not respond to immunotherapy, as compared to responding tumors. However, in contrast to melanoma, macrophages in BCC displayed a low anti-inflammatory gene expression profile, suggesting a minor impact on tumor immunity [20].

#### 2.1.3. Myeloid-Derived Suppressor Cells

Myeloid-derived suppressor cells (MDSCs) are frequently found in the TME of NMSCs [21,22]. This is a heterogeneous population of myeloid cells with immunosuppressive activity, and their presence in tumors is often associated with poor clinical outcome (Figure 1) [23]. In various types of cancer, including melanoma, MDSCs were shown to induce tumor growth and progression by promoting the immunosuppressive TME, e.g., by inhibiting NK cells, CD4^+^ T cells and CD8^+^ T cells, and stimulating the activity of regulatory T cells (Tregs) [24,25]. STAT3 is a transcription factor and known oncogene activated by pro-inflammatory cytokines. In cSCC, STAT3 was shown to stimulate the activation and proliferation of MDSCs but also promote the MDSC-induced switch from TAMs to an immunosuppressive M2 phenotype [26].

#### 2.1.4. Neutrophils

High-risk cSCC displaying markers of increased metastatic risk are associated with high numbers of both tumor-associated neutrophils (TANs) and circulating neutrophils [27]. Neutrophils are an essential part of the innate immune system. Although neutrophils generally have antitumor properties, high numbers of TANs are associated with poor prognosis in many types of cancer [28] (Figure 1). The gene expression analysis of TANs in the DMBA/PMA-induced murine cSCC model revealed a predominant protumor gene expression signature during tumorigenesis, as compared to adjacent skin [29]. This indicates that under the influence of the TME neutrophils shift to a tumor-promoting phenotype. Moreover, in this murine cSCC model, TANs displayed an increased expression of immunosuppressive markers PD-L1, Siglec F, reactive oxygen and nitrite production and PD-L1 expression on TANs correlated to tumor growth. The depletion of TANs delayed tumor growth and restored the antitumor CD8^+^ T cell responses, indicating that PD-L1+ TANs suppress tumor-specific CD8^+^ effector T cells expressing PD-1, through PD-L1/PD-1 signaling [29].

#### 2.1.5. Mast Cells

Mast cells are prevalent in the skin and other peripheral tissues, but their role in cancer in has not been studied in detail. In mice, the levels of dermal mast cells was shown to correlate with their susceptibility to the UV-induced suppression of systemic contact hypersensitivity responses. Studies in mast cell-depleted mice showed that mast cells are required for UV-induced immunosuppression [28,30]. In BCC patients, sun-protected skin showed a more abundant presence of dermal mast cells than in healthy individuals [31], suggesting elevated dermal mast cell level as a predisposing factor for BCC development in humans. A functional connection is suggested by the murine data that dermal mast cells promote UVB-induced immune suppression and thereby decrease cancer immunosurveillance (Figure 1) [28,30].

#### 2.1.6. Innate lymphoid Cells and Natural Killer Cells

Group 1 innate lymphoid cells (ILC1) that lack the expression of antigen-specific receptors and produce inflammatory cytokines IFNγ, TNFα and GM-CSF. Natural killer (NK) cells are a subset of ILC that are able to kill tumor cells by secreting perforins and granzymes [32], while other ILC1 lack cytotoxic properties. The role of NK cells in NMSC surveillance has mostly been studied in murine cSCC models. In the DMBA/PMA-induced murine cSCC model, NK cells were found to accumulate in DMBA-treated skin. NK cell infiltration depended on TNFα produced by Langerhans cells, which in turn activates the epidermal production of chemokines CCL2 and CXCL10. These NK cells expressed NKG2D, by which they can kill transformed cells expressing ligands of NKG2D. This was demonstrated via NK cell depletion experiments that resulted in an accumulation of DNA-damaged cells, and an increased number of papillomas, indicating that NK cells are involved the early elimination of DNA-damaged, transformed keratinocytes [11]. These data show the cooperation between NK cells and Langerhans cells in suppressing cSCC development.

Although less characterized, the role of noncytotoxic ILC1 in antitumor immunity is thought to be in cancer immunoediting [33]. ILC1 thus mediate cancer immunity by producing inflammatory cytokines in the TME that promote local T cell activation. However, GM-CSF production by ILC1 may activate macrophages and facilitate tumor progression. In both precancerous murine cSCC lesions and cSCC tumors and human cSCC, infiltrating NK cells and ILC1 have been shown to display an atypical cytokine secretion profile, impaired cytotoxicity and are possibly incapable of eradicating tumor cells [34]. Since several studies have reported the downregulation of NK cell-activating receptors and the presence of exhaustion markers on tumor-infiltrating NK cells, possibly through IL-33 cytokine signaling, it is likely that the TME can promote the dysfunction of NK cells, thus allowing tumor development [35,36]. In addition, impaired NK function has been associated with an increased risk in cSCC [37]. The role of ILC3 and ILC2 in antitumor immunity may involve macrophage activation and eosinophil recruitment, respectively [33], but this has not yet been studied in cSCC or BCC.

This suggests that besides their antitumor response, NK cells and noncytotoxic ILC1 also play a dual role in cancer immunosurveillance and tumor establishment (Figure 1).

#### 2.1.7. NKT Cells and γδ T Cells

NKT cells express an αβ T cell receptor that recognizes glycolipids bound to CD1d. NKT cells can have both immunosuppressive and stimulatory roles in the skin. In murine models, NKT cells were involved in the UV-induced immunosuppression and permitted outgrowth of UV-induced tumors and HPVE7 oncogene-driven skin tumors [38]. The immune suppressive effect was dependent on NKT cell-derived IFNy that likely stimulates IDO production by IFNy-responsive myeloid cells.

Intraepithelial lymphocytes comprise γδ T cells, a special subset of T cells lacking αβ T cell receptor and CD4 or CD8 coreceptor expression. Their role in skin cancer has not widely been studied. Studies in mice have shown that mice lacking γδ cells are more susceptible to skin carcinogenesis, which might involve γδ T cell-mediated tumor cell killing via NKG2D [39]. However, since γδ T cells are abundantly present in murine skin and only occasionally found in human skin, and the impact of these cells on human skin cancer development is less clear.

### 2.2. Adaptive Immunity against NMSC

#### 2.2.1. T Cell Responses

T cell responses are primed upon antigen presentation by activated dendritic cells in the lymph nodes. CD8^+^ T cells can exert direct antitumor activity by secreting Granzyme B and perforin (Figure 1). CD8^+^ T cells also secrete several cytokines, including TNFα and interferon-γ (IFNγ), which can activate other immune cells [40]. The role of CD8^+^ T cells in controlling cSCC was shown in CD8^+^ T cell knockout mice that developed more cSCC tumors upon UVB irradiation than wild type mice [41]. In a murine model of transplanted SCC that grow under immunosuppression by tacrolimus, it was found that tumor rejection upon tacrolimus withdrawal was dependent on the presence CD8^+^ T cells. This effect was mediated by IFNγ, since neutralizing IFNγ resulted in tumor progression and decreased survival [42]. However, within the TME, tumor-infiltrating CD8^+^ T cells often show an exhausted phenotype, characterized by decreased cytokine production and cytotoxic activity coinciding with the upregulation of inhibitory receptors PD-1 and CTLA-4, which decreases their antitumor activity [43,44].

In a study of BCC patients, patients with recurrent episodes had a significantly lower number of infiltrating CD8^+^ T cells and dendritic cells in the primary tumor than patients without recurrence, suggesting that their involvement in the chance of BCC recurrence [45]. Likewise, progressive head and neck cSCC displayed lower CD8^+^ and CD4^+^ T cell responses and more regulatory T cell (Treg) infiltration in primary tumors than in non-progressing cSCC [46]. CD4^+^ T-helper 1 (Th1) cells are considered to promote antitumor activity by producing cytokine interleukin-2 (IL-2) and IFNγ, which recruit and activate immune cells, and by stimulating the cytotoxic CD8^+^ T cell response. T helper 2 (Th2) cells express IL-4, IL-5 and IL-13, which have been associated with immune tolerance in transplantation. The gene expression analysis of cSSC and the surrounding skin in immunosuppressed, solid organ transplant recipients (SOTRs) showed that reduced CD4 T cell infiltration has a predominant Th2 expression profile, as compared to immuno-competent SSC patients. This reveals that the perineoplastic microenvironment in the adjacent non-tumor-bearing skin of SOTRs differs from immunocompetent individuals in suppressing Th1 responses and favoring Th2 polarization, thereby facilitating more SCC recurrence in SOTRs [47].

T-helper 17 (Th17) cells primarily express IL-17, an inflammatory cytokine that has been implicated in the proliferation of keratinocytes and the development of cSCC in the murine DMBA/TPA-induced cSCC model [48]. IL-17-mediated inflammation was also shown to be required for UVB-induced immunosuppression in the skin by inducing regulatory T cells (Tregs) and tolerogenic myeloid cells [49]. UV-induced skin damage decreases the antigen-presenting function of skin DC and causes Tregs to migrate to the skin, thereby dampening immunosurveillance and indirectly promoting tumor establishment. Tregs suppress the activity of other lymphocytes as immune regulation to limit the damaging effects of prolonged inflammation, whereas in cancer they shape the immunosuppressive TME [50]. Loser et al. (2007) [51] demonstrated that IL-10 knockout mice were protected from skin carcinogenesis after UV radiation. In these mice, the suppressive function of UV-induced Tregs was impaired, resulting in the enhanced antitumor activity of CD4^+^ and CD8^+^ T cell responses. In human cSCC tissues, more Tregs expressing immunosuppressive cytokines, including IL-10, and promoting T cell exhaustion were found in poorly differentiated G2–G3 stages than in well-differentiated G1 stages, indicating their correlation to tumor aggressiveness [52].

Th17 cells also secrete IL-23, which promotes chronic inflammation within the TME, which, in turn, results in increased tumorigenesis in the DMBA/TPA chemical murine skin carcinogenesis model [53]. In contrast to IL-12, which promotes antitumor immunity, IL-23 was shown to induce an inflammatory response, characterized by MMP9 expression and increased angiogenesis, while reducing CD8^+^ T cell infiltration. This effect could be reversed by the elimination of IL-23, resulting in protection against tumor development [53].

#### 2.2.2. B Cell Responses

The role of B cells in cancer is not fully clear and sometimes even contradictory. B cells have been shown to both inhibit tumor growth by promoting NK cells and macrophages but also to stimulate tumors by secreting tumor growth factors. The underlying mechanisms that drive B cells to be either anti-tumor or pro-tumor are not fully understood [54]. In cancer, vascularized tertiary lymphoid structures (TLS) comprising B cells, DC and T cells can arise in non-lymphoid tissues. TLS density has been associated with improved response to immunotherapy in various cancer types [55]. In BCC, TLS were more abundantly found in the nodular BCC subtype than in BCC without a nodular component, and more mature TLS numbers were associated with increased tumor-infiltrating T cell levels and tumor cell killing [56]. Moreover, the presence of memory B cells was correlated with a response of BCC to immunotherapy and was negatively correlated to macrophage presence. The balance between the antitumor activity of B cells and the inhibition of B cells by macrophages determines immunotherapy responsiveness [20].

High numbers of memory B cells in the peritumoral stroma were also associated with the improved progression-free survival of cSCC patients [57]. In a murine multistage cSSC model of HPV16 mice, de Visser et al. (2005) [58] demonstrated that chronic inflammation promotes epithelial hyperproliferation, tissue remodeling and angiogenesis during premalignancy. The local deposition of immunoglobulin was observed during chronic skin inflammation even though B cells did not infiltrate the premalignant skin. Through the adoptive transfer of B cells or serum into immunodeficient HPV16/RAG-1 knock out mice, they showed that peripheral B cell activation and B cell-derived factors, including antibodies, play a major role in driving chronic inflammation associated with carcinogenesis [58]. B cells were also described to promote TNFα-dependent carcinogenesis in the murine DMBA/TPA-induced cSCC model. This effect was mediated by regulatory B cells that produce TNFα and IL-10 and suppress autoimmune Th1 responses during chronic inflammation. As a result, malignant cells arising during chronic inflammation are not effectively eliminated [59]. The pro-tumorigenic role of B cells during skin carcinogenesis was further demonstrated in a murine model of UV-induced cSCC [60]. B cell depletion prior to tumor establishment did not affect tumor development, but the absence of B cells in the established tumor phase diminished tumor growth and metastasis.

### 2.3. Increased Risk of Non-Melanoma Skin Cancer (NMSC) in Solid Organ Transplant Recipients (SOTRs)

The increased incidence of BCC and cSCC in SOTRs emphasizes the importance of immunosurveillance and effective immune responses in skin cancer prevention. cSCC and BCC are responsible of more than 95% of the cancers found in SOTRs [61]. The NMSC incidence in SOTRs within 5 years varies from 1.5 to 22% for liver transplants, 2 to 24% for kidney transplants and 6 to 34% for heart transplant recipients, depending on the geographic location and other factors [62]. One of the factors that contribute to the high incidence rate of BCC and cSCC in SOTRs is the prolonged use of immunosuppressive medication to prevent graft rejection. Interestingly, whereas the BCC:SCC incidence ratio in immunocompetent individuals is 4:1, this is reversed in SOTRs, ranging from 1:2.4 to 1:3.8 depending on the geographic location [62]. Recipients of an organ have a 153-fold increased risk of developing cSCC in their life as compared to immunocompetent people, which is even higher at a younger age (480-fold increased risk, age < 50 years) [61]. Incidence ratios of BCC have not exactly been estimated due to a lack of comparable data from the general population. Although recent research shows a declining cSCC incidence in SOTRs, possibly due to a combination of a lower dosage of immunosuppressive medication given and better surveillance [63], the cSCC risk remains significantly increased. Moreover, cSCC tumors in SOTRs are often more aggressive and malignant, making it the most common cause of death in SOTRs [64]. Interestingly, the tumor genetic profiles of cSCC in SOTRs did not significantly differ from the cSCC of immunocompetent patients, suggesting that the more aggressive clinical course of cSCC in SOTRs is mainly due to impaired antitumor immunosurveillance and/or antitumor immunity rather than tumor genomic factors [57].

The class of medication patients receive greatly influences the risk of developing cSCC and BCC. Here, we describe the main types of immunosuppressive medication applied in SOTRs and how they are able to affect skin cancer susceptibility (Table 1). This analysis also reveals which immune mechanisms are important for cSCC and BCC prevention.

#### 2.3.1. Calcineurin Inhibitors

Calcineurin inhibitors consist of cyclosporin A (CsA) and tacrolimus. Tacrolimus was first approved 1994 as an effective replacement of CsA. Since then, both tacrolimus and CsA are the main immunosuppressive drugs used after organ transplantation. CsA has been associated with the increased incidence of cSCC in OTRs [2]. Although reports on tacrolimus have been contradictory, it seems that the risk of cSCC does not differ between tacrolimus and CsA users [2].

Calcineurin inhibitors mediate immunosuppression via the NFAT/calcineurin pathway. They bind to cyclophilins in T lymphocytes, which block calcineurin activity. Normally, calcineurin dephosphorylates NFAT proteins, leading to the transcription of the IL-2 gene. Calcineurin inhibition leads to the inhibition of IL-2 production and the decreased activation and proliferation of T cells [66], as well as the downstream activation of cellular and humoral immunity. CsA has been shown to affect multiple NFAT family members, namely NFAT1, NFAT2 and NFAT4, resulting in the inhibition of IL-2, IL-4 and CD40L [66]. Other transcription factors involved in the IL-2 gene transcription, i.e. AP-1 and NF-kB, are also decreased by CsA, possibly through the inhibition of the JNK and p38 pathways. This indicates that CsA, and probably also tacrolimus, have multiple targets to mediate immune suppression.

Early data suggest that the inhibition of T cell-derived IFNγ by calcineurin inhibitors decreases monocyte activation and IL-1 production as well as the downstream function of innate immunity in activating helper T cells responses [86]. Ohata et al. (2011) [71] showed that both calcineurin inhibitors can inhibit the proliferation of NK cells. This coincided with an increase in CD16^-^CD56^bright^ NK cells, mainly expressing cytokines, whereas the CD56^dim^ NK cells are mostly responsible for cytotoxic activity and the killing of target cells. Tacrolimus has been shown to inhibit the CXCL10 and IL-12 production by human monocyte-derived DCs in vitro, which impairs their ability to prime T cells [76]. Although DC and Langerhans cells found in human cSSC tissues displayed a mature phenotype, they appeared to be poor stimulators of T cell responses as compared to DCs from adjacent non-tumoral skin [13]. This shows the suppressive influence of the TME on DC function. In murine bone marrow-derived DCs, both CsA and tacrolimus were shown to block intracellular MHC-I antigen processing and presentation [65], whereas MHC expression and phagocytic activity was unaffected. This indicates that CsA and tacrolimus can enhance the risk of cSCC development by impairing both the innate immunity and antigen presentation of DCs resulting in decreased adaptive immunosurveillance.

CsA induces the production of TGFβ, which has been associated with the nephrotoxicity of CSA treatment and may promote cancer progression [66]. In murine cSCC, the effect of TGFβ1, being either tumor suppressive or tumor promoting, has been shown to depend on the stage of tumorigenesis [69]. As TGFβ1 inhibits the proliferation of epithelial cells, TGFβ1 overexpression prior to tumor formation suppressed benign papilloma formation in transgenic mice [67]. On the other hand, TGFβ1 overexpression after papilloma establishment stimulated malignant transformation and metastasis [68]. Increased TGFβ1 expression has been found in patients cSCC tissues [70]. Murine studies of inducing sustained TGFβ levels, comparable to those found in human cSSC, resulted in epithelial hyperplasia, inflammation and increased angiogenesis [70]. Inflammation was predominantly characterized by granulocytic infiltration and proinflammatory cytokines, IL-1β and TNFα, suggesting that the chemotactic effect of TGFβ1 overruled its anti-inflammatory effect.

Zhang et al. (2013) [72] found that the cSCC TME of SOTRs receiving calcineurin inhibitors (frequently combined with azathioprine and mycophenolate mofetil) has a different T cell polarization profile than immunocompetent patients. Increased Treg numbers and Treg/CD8^+^ T cells ratios and lower IFNγ-producing CD4^+^ T cell numbers indicated an immunosuppressive tumor environment in cSCC from SOTRs. Furthermore, they found an increase in the numbers of IL-22-producing CD8^+^ T cells and IL-22 and IL-22 receptor (IL-22R) expression and proliferation markers. Although IL-22 is known to promote inflammation as well as inhibit apoptosis of keratinocytes [87], it induced the proliferation and invasiveness of the human cSCC cell line A431 in vitro [72]. The in vitro treatment of A431 cells with CsA increased IL-22R expression, indicating that CsA renders SCC cells more sensitive to IL-22. Likewise, in UVB-induced SCC in immunocompetent mice, CsA was shown to drive T cell polarization towards an IL-22 response and to increase IL-22R expression on SCC and their invasive capacity [73]. This effect was reversed via treatment with the anti-IL-22 antibody, which decreased the tumor number and burden [73]. These findings are in line with SCCs found in SOTRs, being more aggressive than in immunocompetent patients [88].

Besides immune suppression, CsA might also directly promote tumor development by inhibiting NFAT expressed in epidermal keratinocytes. In immunocompromised mice, blocking NFAT by CsA resulted in the decreased expression of P53, an important tumor-suppressor gene, and increased keratinocyte tumor formation [74]. NFAT is a negative regulator of oncogene ATF3, an Ap-1 family member. CsA treatment led to ATF3 upregulation and suppression of cancer cell senescence, both in human skin explants and tumor xenografts, which was reversed through ATF3 knockdown [74]. Likewise, ATF3 was upregulated in cSCC tissues of immunosuppressed patients, as compared immunocompetent patients [75]. Moreover, the upregulation of ATF3 in human keratinocytes by CsA was potentiated via UVA irradiation, through ROS and nuclear factor erythroid 2–related factor 2 (NRF2) activation [75]. This indicates that whereas CsA treatment is systemic in SOTRs, the combined effect of CsA and UV exposure renders the skin more at risk for tumor development than other organs.

#### 2.3.2. Purine Analogues

Azathioprine (AZA, Imuran) and mycophenolate mofetil (MMF) belong to the purine analogues, and both inhibit DNA and RNA synthesis. AZA inhibits purine synthesis, required for DNA synthesis. MMF is a prodrug of mycophenolic acid (MPA), which inhibits inosine monophosphate dehydrogenase (IMPDH), a rate-limiting enzyme in DNA and RNA synthesis [71,77]. Blocking DNA and RNA synthesis affects actively proliferating cells, such as T and B lymphocytes. AZA and MMF have been associated with the increased incidence of cSCC in SOTRs, although the association is stronger with AZA [2].

Besides the inhibition of T and B cell proliferation, MMF can also downregulate the expression of lymphocyte adhesion molecules, including vascular cell adhesion molecule 1 (VCAM-1), and inhibit the recruitment and migration of both monocytes and lymphocytes [80].

Immunosuppression by AZA is mediated by inhibiting the activation of Rac1 in CD4^+^ T cells, which leads to the downregulation of anti-apoptotic protein Bcl-xL, resulting in the increased apoptosis of T lymphocytes [78]. AZA also inhibits the expression of pro-inflammatory genes TRAIL, TNFRS7 and a4-integrin in activated T cells [89], thereby reducing their effector function. Since CD4^+^ T cell responses are crucial for effective CD8^+^ T cell immunity, the induction of CD4^+^ T cell apoptosis by AZA can increase the risk of developing cSCC.

MMF also affects innate immunity and has been shown to inhibit the proliferation of NK cells in vitro even more effectively than CsA and TAC [71]. Interestingly, while CsA and TAC increased the proportion of CD16^−^CD56^bright^ NK cells, MMF decreased the proportion of this subset. Moreover, MMF also reduced the expression of activating NK receptors and NK cell cytotoxicity [71]. Impaired NK function has also been associated with an increased risk of cSCC [37], suggesting its role in the early response against cSCC [11]. Thus, NK cells can play a crucial role both in immunosurveillance to mediate tumor prevention and in the antitumor response against established tumors.

Besides immunosuppression, AZA may also promote skin cancer in a non-immunological manner. AZA increases the sensitivity of the skin and promotes the accumulation of 6-thioguanine in the DNA of patients. When exposed to UVA irradiation, this can lead to the production of carcinogenic reactive oxygen species (ROS) that promote mutagenesis [79].

#### 2.3.3. mTOR Inhibitors

Another class of immunosuppressive medication that was recently introduced are mTOR inhibitors, consisting of sirolimus, also known as rapamycin, and everolimus. mTOR inhibitors are associated with a lower risk of developing de novo malignancies or secondary cSCC in SOTRs than calcineurin inhibitors and purine analogues [4,90]. Similar to tacrolimus, sirolimus binds to FK-binding protein 12 (FKBP12), but the downstream effect is different. While the tacrolimus-FKBP12 complex inhibits calcineurin, the sirolimus-FKBP12 complex inhibits mTOR signaling by binding to mTOR Complex 1 (mTORC1). Everolimus is a rapamycin analog with a higher selectivity for mTORC1 complex binding than sirolimus. By inhibiting mTOR signaling, ribosomal p70S6 kinase is inactivated, which results in the downregulation of genes involved in cell cycle phase shifting from G_1_ to S. Thus, mTOR inhibitors block lymphocyte proliferation induced by IL-2 signaling but do not interfere with IL-2 production itself [4].

However, rapamycin has been shown to exert immunostimulatory effects during infection by regulating memory T cell formation [91]. Rapamycin increased the number of virus-specific CD8^+^ T cells and stimulated differentiation into memory CD8^+^ T cells in mice infected with lymphocytic choriomeningitis virus (LCMV) and in nonhuman primates vaccinated with modified vaccinia virus [91]. This effect also translates to immunosuppression during transplantation and cSSC risk, as shown by Jung et al. [82]. Rapamycin suppressed the rejection of CD8^+^ T cell-mediated skin grafts in mice, demonstrating its immunosuppressive effect. Interestingly, the long-term rapamycin treatment of K14 HPV38 E6/E7 transgenic mice with UV-induced actinic keratosis (AK) and cSCC lesions showed increased differentiation and enhanced CD8^+^ memory T cell function in the skin. The infiltration of CD8^+^ effector memory T cells into the AK and cSCC lesions was also increased upon rapamycin treatment, as compared to mice treated with tacrolimus. In a long-term contact hypersensitivity model, it was shown that this effect of rapamycin only occurred when rapamycin was present during the sensitization phase, indicating that this effect is restricted to new antigenic challenges [82]. These findings suggest that SOTRs treated with sirolimus might benefit from increased CD8^+^ memory T cell function, which might lower the risk of cSCC development or progression. Enhanced CD8^+^ memory T cell function might also explain the regression and reduced recurrence rate of cSCC in patients who switched from calcineurin inhibitors to sirolimus [90,92].

mTOR inhibitors have divergent effects on DCs differentiation and maturation, depending on the type of DC analyzed, myeloid DCs (mDCs) or monocyte-derived DCs (moDCs), and the antigenic stimulus [81]. Rapamycin induced the apoptosis of DC in moDCs cultures by interfering with GM-CSF signaling but did not affect freshly isolated monocytes, macrophages or myeloid cells [93]. The sustained mTOR inhibition of moDCs decreased the expression of costimulatory molecules and inflammatory cytokines, which is indicative of immunosuppression. In mDCs activated by TLR or TLR-independent stimuli, rapamycin induced the increased expression of NF-kB and several other pro-inflammatory cytokines, while anti-inflammatory IL-10 and STAT3 were downregulated. Moreover, the mDCs of kidney transplant patients treated with sirolimus more effectively stimulated T cells than patients treated with calcineurin inhibitors, whereas mDC differentiation was not affected in rapamycin-treated patients [81]. As described above, in cSCC, STAT3 can function as an oncogene that activates MDSCs and induces a phenotypical switch from pro-inflammatory M1 to immunosuppressive M2 macrophages. Therefore, sirolimus might prevent the reduced immunosurveillance and constitutive development of cSCC via the downregulation of IL-10 and STAT3 and the upregulation of pro-inflammatory cytokines. Unlike CsA and tacrolimus, which block intracellular MHC-I antigen processing and presentation in dendritic cells, this pathway is not influenced by mTOR inhibitors [65]. Therefore, patients receiving mTOR inhibitors likely have intact immunosurveillance by DCs to prime an antitumor T cell response.

Besides impacting on immunologic responses, mTOR inhibitors also have anti-neoplastic and anti-proliferative action. Since many different tumors, including renal and breast carcinoma, induce mTOR dysregulation, the disruption of this pathway can inhibit tumor growth and sometimes even lead to tumor regression [83]. Concerning skin cancer, DeTemple et al. [84] demonstrated that the in vitro treatment of human keratinocytes with sirolimus and everolimus resulted in the upregulation of IL-6. This resulted in downregulation in cytokeratin 10 (CK10), a protein implicated in determining epidermal thickness. CK10 knockout mice treated with DMBA, a chemical carcinogenic substance, developed fewer tumors and reduced tumor formation compared to control mice [85]. Thus, inhibiting mTOR leads to the IL-6-mediated downregulation of CK10, a protein that might play an important role in the tumorigenesis of keratinocytes. Despite these anti-tumor effects, mTOR inhibitors are rarely given immediately after solid organ transplantation, because they have been associated with the increased incidence of wound healing complications and severe side effects, including peripheral edema and hypertriglyceridemia [94,95].

## 3. Discussion

The involvement of the immune system in the development and progression of cSCC, and to a lesser extent BCC, has been well established, even though the exact underlying mechanisms remain unclear. The inverse ratio of BCC to cSCC, resulting from the higher prevalence of cSCC in SOTRs, may indicate a relatively larger role of immunosurveillance in the prevention of cSCC than BCC. The UV-induced gene mutations, found in cSCC, commonly occur in sun-exposed skin, indicating that immunosurveillance is crucial to prevent these mutated cells from undergoing malignant transformation. Moreover, the increased mutational load in UV-exposed skin cells enhances the frequency of neoantigens that can be recognized by the adaptive immune system, thereby empowering antitumor immunity.

Most knowledge of antitumor immunity in NMSC is based on cSCC, being the most prevalent tumor in SOTRs, and since the preclinical models of chemical or UV-induced skin cancer generally involve cSCC development. However, cSCC and BCC are tumors of different pathogeneses and clinical behaviors, which justifies further studies to explore immunity against BCC in more detail.

Research conducted in preclinical models and immunocompromised patients has indicated the importance of immunosurveillance and both innate and adaptive immune activation. As first line of defense, innate immune cells, NK cells and DCs, play an important role in tumor recognition and eradication as well as in activating and recruiting other immune cell types. UV-induced damage to DCs and NK cells can inhibit their function and allows for tumor promotion. Adaptive immunity in NMSC balances between CD8^+^ T cell responses with direct antitumor activity and immune regulation/suppression through (UV-damaged) DCs, Th2 or Th17 responses that induce Treg activity. In established tumors, mast cells, MDSC, TAMs and TANs contribute to the immunosuppressive environment of the TME and interfere with the adaptive immune response. B cells can also contribute to skin carcinogenesis by promoting chronic inflammation or regulatory B cell activity.

Transplant recipients have a significantly higher risk of developing sSCC and BCC than immunocompetent individuals, depending on the class of immunosuppressive medication. In particular, calcineurin inhibitor CsA and purine analogue AZA—and to a lesser extent, tacrolimus and MMF—are associated with increased cSCC incidence. mTOR inhibitors sirolimus and everolimus have a lower cSCC risk than calcineurin inhibitors and purine analogues, which reflects the different immunological targets of these compounds. In addition to immune suppression, the cSCC risk in SOTRs is affected by the balance between the tumor-promoting or -suppressing effects of different immunosuppressive drugs. Table 1 summarizes the immunological and non-immunological effects of the different immunosuppressants.

Regarding immune suppression, calcineurin inhibitors act primarily by inhibiting the proliferation of T cells via the NFAT/calcineurin pathway, leading to IL-2 production. They also impair T cell priming by blocking MHC-I antigen processing and presentation by DCs or drive T cell polarization towards an IL-22 response. In addition, this class of drugs can also inhibit the proliferation of NK cells and promote a less cytotoxic phenotype. Importantly, calcineurin inhibitors seems to disrupt immunosurveillance in the skin by inhibiting DC and NK function and impair adaptive antitumor T cell responses.

Purine analogues also inhibit the proliferation of lymphocytes and the recruitment of monocytes and lymphocytes to the skin. Furthermore, AZA is also able to induce apoptosis in CD4^+^ T cells, resulting in insufficient T cell help to activate CD8^+^ T cell immunity. MMF inhibited the proliferation of NK cells more strongly than CsA and tacrolimus, and decreased the activation of NK receptors and cytotoxicity. Taken together, purine analogues predominantly interfere with the expansion of adaptive and innate immune responses.

Like calcineurin inhibitors, mTOR inhibitors block lymphocyte proliferation by interfering with IL-2. However, CsA and tacrolimus inhibit the production of IL-2, while mTOR inhibitors block IL-2 signaling. The main difference between these types of compounds with regard to cancer risk may result from the fact that, unlike calcineurin inhibitors, mTOR inhibitors do not interfere with the intracellular antigen processing and presentation of mDCs. Moreover, mTOR inhibitors even improve the T cell-stimulating ability of mDCs in kidney transplant recipients. Rapamycin also promotes the differentiation and infiltration of memory CD8^+^ T cells into the skin and cSCC lesions. These immunostimulatory effects of mTOR inhibitors might explain the reduced recurrence of cSCC in SOTRs who switched from calcineurin inhibitors to rapamycin treatment. In conclusion, mTOR inhibitors seem to promote immune suppression by blocking lymphocyte proliferation. In contrast to calcineurin inhibitors and purine analogues, during mTOR inhibitor treatment, key immune cells that are involved in immunosurveillance remain intact.

Immunosuppressive drug classes greatly differ in their direct effects on skin carcinogenesis and tumor growth. AZA can promote tumorigenesis by sensitizing the skin to UV irradiation and increasing oxidative stress in the skin. Blocking calcineurin signaling via CsA can induce the NFAT-dependent inhibition of the tumor-suppressor gene P53 in keratinocytes. The tumor-promoting characteristics of tacrolimus and MMF have not been identified so far, which might (partially) explain the lower cSCC incidence of tacrolimus and MMF than AZA and CsA. Conversely, mTOR inhibitors may exert antitumor activity. Inhibiting mTOR signaling decreases the proliferation of tumor cells or tumor regression, possibly by decreasing epidermal thickness through the downregulation of CK10.

Considering all these findings, it seems that sirolimus/rapamycin is able to avert cSCC development by keeping intact responses that are involved in immunosurveillance and the activation of the adaptive immune system. In contrast, immune suppression through calcineurin inhibitors and purine analogues mainly seem to focus on inhibiting the function of monocytes, DCs and NK cells, indicating that these cells might have an important function in the immunosurveillance and prevention of NMSC development. Despite the distinct immunological mechanisms that are targeted by these immunosuppressants, it is unlikely that the increased incidence of NMSC in SOTRs can be explained solely by immunosuppression. NMSC incidence associated with the immunosuppressive regimen likely results from the balance between immunosuppression and the direct effects on tumor formation and growth. Further studies focusing on unwinding the immunosuppressive and (anti)mutagenic effects of these drugs may improve the current immunosuppressive regimens in SOTRs.

## Figures and Tables

**Figure 1 cells-12-02441-f001:**
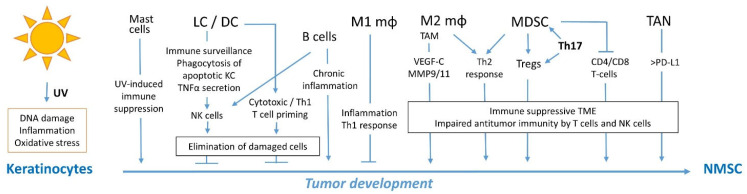
Overview of immune cells involved in immunosurveillance and immunity against NMSC. Abbreviations, LC, Langerhans cells; DC, dendritic cells; Mφ, macrophages; TAM, tumor-associated macrophages; Tregs, regulatory T cells; TAN, tumor-associated neutrophils; TME, tumor microenvironment.

**Table 1 cells-12-02441-t001:** Immunosuppressive drugs taken by solid organ transplants and their effect on immune cells and their non-immunologic effects on tumorigenesis. CsA = cyclosporin A. TAC = tacrolimus. AZA = azathioprine. MMF = mycophenolate mofetil. SIR = sirolimus/rapamycin. EV = everolimus.

Drugs	DCs	Myeloid Cells	NK Cells	Lymphocytes	Non-Immunologic Effects on Tumorigenesis
*Calcineurin inhibitors*					
CsA	Impairs MHC-I antigen processing and presentation [65]	Induction of TGFβ1 production, which suppresses early tumor formation but also stimulates malignant transformation and metastasis of established tumors [66,67,68,69] Induction of granulocytic inflammation and proinflammatory cytokines IL-1β and TNFα [70]	Inhibition of NK cell proliferation; increases the proportion of CD16^−^CD56^bright^NK cells, which are not cytotoxic but express IL-10 and IL-13 [71]	Dephosphorylation of NFAT family members NFAT1, NFAT2 and NFAT4, resulting in the inhibition of IL-2 and IL-4 production and the decreased activation and proliferation of T cells [66] Increased Treg/CD8^+^ T cell ratios; lower IFNy-producing CD4^+^ T cells numbers [72] Induction of Th22 response [72,73]	Downregulation of NFAT in keratinocytes decreases the expression of tumor suppressor gene P53 [74], the carcinogenic effect of which is potentiated by UV exposure [75]
TAC	Inhibition of CXCL10 and IL-12 production by DC, impairing their T cell priming ability [76] Impairs MHC-I antigen processing and presentation [65]	-	Inhibition of NK cell proliferation; increases the proportion of CD16^−^CD56^bright^NK cells, which are not cytotoxic but express IL-10 and IL-13 [71]	Dephosphorylation of NFAT, resulting in the inhibition of IL-2 and IL-4 production, decreased the activation and proliferation of T cells [66]	-
*Purine analogues*					
AZA	-		-	Inhibition of DNA and RNA synthesis, resulting in the suppression of lymphocyte proliferation [71,77] Downregulation of Bcl-xL, resulting in increased apoptosis in CD4^+^ T cells Inhibition of pro-inflammatory gene expression [77,78]	Increases the photosensitivity of the skin. Promotes the accumulation of 6-thioguanine in the DNA, leading to increased oxidative stress and mutagenesis upon UV irradiation [79]
MMF	-	-	Inhibition of NK cell proliferation; decreases the proportion of CD16^−^CD56^bright^NK cells, which are anti-inflammatory [71]. Downregulation of activating NK cell receptors and NK cell cytotoxicity [71]	Inhibition of DNA and RNA synthesis, resulting in the suppression of lymphocyte proliferation [71,77] Downregulation of VCAM-1 expression and the inhibition of recruitment and the migration of lymphocytes [77,80].	-
*mTOR inhibitors*					
SIR	Decreased expression of costimulatory molecules and inflammatory cytokines by moDC Increased expression of NF-kB and other pro-inflammatory cytokines by mDC upon stimulation, and the downregulation of IL-10 and oncogene STAT3 [81]	Prevention of reduced immunosurveillance and constitutive development of cSCC via the downregulation of IL-10 and STAT3 and the upregulation of pro-inflammatory cytokines [81]	-	Binding to FKBP12, which inhibits mTOR signaling, leading to the inhibition of IL-2 signaling and T cell proliferation. Increased differentiation and enhanced CD8^+^ memory T cell function in the skin, against new antigenic challenges [82]	Anti-proliferative and anti-neoplastic activity [83] Upregulation of cytokine IL-6, resulting in the downregulation of CK10 [84] and less skin tumor formation [85]
EV	-	-	-	Binding to mTOR complex 1, leading to the inhibition of IL-2 signaling and T cell proliferation [4]	Anti-proliferative and anti-neoplastic activity [83] Upregulation of cytokine IL-6, resulting in the downregulation of CK10 [84] and less skin tumor formation [85]

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
