# Peer review of "Immunity against Non-Melanoma Skin Cancer and the Effect of Immunosuppressive Medication on Non-Melanoma Skin Cancer Risk in Solid Organ Transplant Recipients"

_cells, 2023, doi:10.3390/cells12202441_

Round 1

Reviewer 1 Report

Bakker and colleagues present a review addressing the predominantly immunologic mechanisms that (may) underlie the increased risk of non-melanoma skin cancer (NMSC) occurrence in individuals immunosuppressed with drugs for organ transplantation.

The authors use the term NMSC to group together cutaneous squamous cell carcinomas (cSCCs) and basal cell carcinomas (BCCs). However, it is the case that in organ transplant patients the risk of occurrence for cSCCs is considerably higher than for BCCs. As a reader of such a review, I would expect to be provided more detailed pathomechanistic information on the underlying differences. In addition, the authors should consider whether they want to use the (not always appropriate but certainly established) collective term NMSC or whether they would prefer to speak of cSCC and BCC in a more differentiated way.

Overall, the authors discuss a large number of granular data on a wide variety of cell types and interactions, without weighting which events are now the most relevant. One would wish to have a graphical representation (or several: one each for CsA/tacrolimus, purine analogues and mTOR inhibitors) in front of one's eyes that clearly summarizes the complex relationships.

Minor Points:

Table 1 should be presented in landscape format to improve clarity and readability.

Chronic lymphatic leukemia is known to increase the risk for and to worsen the course and prognosis of cSCCs. Are similar mechanisms present in this condition as in cSCCs promoted by drug-mediated immunosuppression?

The authors might consider to provide more clinical data on the incidences of BCCs and cSCCs in organ transplant recipients. They should clarify the temporal dynamics of tumor occurrence after the onset of immunosuppression, but also the dependence on its intensity and the type of organ transplanted. This could be elaborated, for example, in section 2.3 of the manuscript.

Author Response

We thank the reviewer for the comments and we revised the manuscript accordingly:

  • The term NMSC is replaced by the appropriate term cSCC or BCC in the  manuscript.
  • More pathomechanistic information on the underlying differences between cSCC and BCC is added.
  • Figure 1 providing an overview of the immune cells involved in NMSC immunity is added, based on the comments of reveiwer 2. The reviewer 1 suggests to also include a figure of the effects of immunosuppressive drugs on NSMC immunity. We noticed that the Format of Table 1 was shifted to portrait during submission. This has now been reset to landscape, and provides an overview of the effects of immunosuppresive durg classes on immune cells. We feel that a figure would contain similar content and become redundant. Therefore we chose to confine this information to Table 1 and not adding a figure 2.
  • The notion on CLL and worsening of cSCC is interesting and is likely the result of immune suppression by CLL and B cell depletion in CLL therapy. The effect of B cell depletion onf cSCC has been addressed in the paragraph on B cells in revised version. Further elaboration to CLL would go beyond the scope of the current review. 
  • More clinical data on cSCC and BCC in SOTR has been added in paragraph 2.3, as well as the timing of tumor occurrence after the onset of immunosuppression, and the type of organ transplanted. 

Reviewer 2 Report

This review article covers immunity against NMSC and the effects of immunosuppressive medications on NMSC development.  While the manuscript is well written, the breadth of the review is very wide and there is a danger that the literature is underrepresented given that the involvement of each immune cell in NMSC could be the subject of individual reviews.  Detailed comments to help improve the manuscript are as follows:

(1) Line 58 – “Although pDC are usually not found in the skin…” – is this for normal or diseased skin or both?

(2) Line 69 – blocking apoptosis results in aggravated skin inflammation and yet the conclusion is that LCs need to take up apoptotic keratinocytes for this inflammation.  This conclusion needs clarification.   

(3) Line 109 – MDSCs are frequently found in the TME of NMSC – need a reference for this statement?

(4) The potential role of NKT cells in skin tumours is not included (e.g.  McKee et al. (2014) J Leuk. Biol. 96, 49-54).  In addition, gammadelta T cells are not mentioned (eg. Girardi et. al. (2001) Science 294:605-609).  Beyond NK cells, the authors may also want to consider other ILCs in skin cancer (eg. Azin et. al. (2021) J. Invest. Dermatol. 141: 2320-2322.)  

(5) Role of B cells – there is more literature that could be cited here (for example, Kok et. al. (2020) J. Invest. Dermatol. 140:1459-1463).

(6) The authors should consider a diagram which summarises the immune cell roles in skin tumour development.

(7) The authors highlight the production of IL-2 as being a key difference between mTOR inhibitors and calcineurin inhibitors and yet there is very little speculation in the discussion about how this could lead to different tumour outcomes particularly given that mTOR inhibitors block subsequent IL-2 signalling. 

Quality of English is good.  Only minor checks needed. 

Author Response

We thank the reviewer for the comments and we revised the manuscript accordingly:

  1. statement has been adjusted to normal skin
  2. conclusion has been clarified
  3. Two references have been added
  4. more text and references describing the involvement of ILC types has been added to 2.1.6; an additional paragraph 2.1.7 has been added describing NKT cells and gamma deltaT cells
  5. The paragrah on B cells has been extended to describe their involvement more extensively in both cSCC and BCC, as well as the findings on TLS in skin cancer, and 5 extra references, including those suggested, have been added. 
  6. Figure 1 has been added which provides an overview of the immune cell roles in skin tumour development.
  7. The discussion has been revised discussing the difference in IL-2, but indicating the differential effect on antigen presentation as being a key difference between mTOR inhibitors and calcineurin inhibitors. Moreover, more discussion has been added on the reversed cSCC: BCC ratio in SOTR as compared to immunocompetent individuals. 
  8. Text has been checked to correct minor English typo's

In addtion to the comments, we added more recent information on macrophages in BCC and their relation to B cells. We also discussed recent genomis data on cSSC from SOTR as compared to immunocompetent patients, suggesting that the more aggressive clinical course of cSCC in SOTR is mainly due to impaired antitumor immunosruveillance and/or antitumor immunity rather than tumor genomic factors.